# Insights into Neonatal Cerebral Autoregulation by Blood Pressure Monitoring and Cerebral Tissue Oxygenation: A Qualitative Systematic Review

**DOI:** 10.3390/children10081304

**Published:** 2023-07-28

**Authors:** Daniel Pfurtscheller, Nariae Baik-Schneditz, Bernhard Schwaberger, Berndt Urlesberger, Gerhard Pichler

**Affiliations:** 1Division of Neonatology, Department of Pediatrics and Adolescent Medicine, Medical University of Graz, 8036 Graz, Austria; daniel.pfurtscheller@medunigraz.at (D.P.); nariae.baik@medunigraz.at (N.B.-S.); bernhard.schwaberger@medunigraz.at (B.S.); berndt.urlesberger@medunigraz.at (B.U.); 2Research Unit for Neonatal Micro- and Macrocirculation, Department of Pediatrics and Adolescent Medicine, Medical University of Graz, 8036 Graz, Austria; 3Research Unit for Cerebral Development and Oximetry, Division of Neonatology, Medical University of Graz, 8036 Graz, Austria

**Keywords:** neonate, blood pressure, NIRS, cerebral oxygenation, cerebral autoregulation

## Abstract

Objective: The aim of this qualitative systematic review was to identify publications on blood pressure monitoring in combination with cerebral tissue oxygenation monitoring during the first week after birth focusing on cerebral autoregulation. Methods: A systematic search was performed on PubMed. The following search terms were used: infants/newborn/neonates, blood pressure/systolic/diastolic/mean/MAP/SAP/DAP, near-infrared spectroscopy, oxygenation/saturation/oxygen, and brain/cerebral. Additional studies were identified by a manual search of references in the retrieved studies and reviews. Only human studies were included. Results: Thirty-one studies focused on preterm neonates, while five included preterm and term neonates. In stable term neonates, intact cerebral autoregulation was shown by combining cerebral tissue oxygenation and blood pressure during immediate transition, while impaired autoregulation was observed in preterm neonates with respiratory support. Within the first 24 h, stable preterm neonates had reduced cerebral tissue oxygenation with intact cerebral autoregulation, while sick neonates showed a higher prevalence of impaired autoregulation. Further cardio-circulatory treatment had a limited effect on cerebral autoregulation. Impaired autoregulation, with dependency on blood pressure and cerebral tissue oxygenation, increased the risk of intraventricular hemorrhage and abnormal neurodevelopmental outcomes. Conclusions: Integrating blood pressure monitoring with cerebral tissue oxygenation measurements has the potential to improve treatment decisions and optimizes neurodevelopmental outcomes in high-risk neonates.

## 1. Introduction

The transition period from fetal to neonatal life is associated with significant physiological changes affecting all vital organ systems [1]. Intrauterine, most of the blood bypasses the lungs through the ductus arteriosus due to elevated pulmonary resistance. Immediately after clamping the umbilical cord, there is a significant reduction in the preload of the heart, as up to 50% of the preload is delivered by the placenta. In lambs, it has been shown that this may result in reduction of cardiac output, which may trigger bradycardia [1,2]. With aeration of the lungs immediately after birth, the pulmonary vascular resistance drops, and the pulmonary blood flow increases, leading to an increase in cardiac output [1,2]. Most changes from the fetal to neonatal transition occur within the first few minutes, which is one of the most-challenging periods in human life [3,4]. Once the immediate transition is completed, cardio-circulatory and pulmonary changes still continue for several weeks [4]. It takes up to 96 h for term infants to have a functional closure of the ductus arteriosus. Permanent anatomic closure occurs within three weeks up to three months after birth [4]. Therefore, monitoring to assess and evaluate hemodynamics in neonates, especially during the first week after birth, is crucial, whereby the immediate transition might also be highly relevant.

During immediate transition after birth, monitoring with pulse oximetry and electrocardiogram (ECG) is recommended [5,6,7]. However, the routinely used monitoring is not always sufficient to recognize potentially compromised neonates [8], especially in cases when impaired cardio-circulation causes reduced oxygen delivery to the tissue.

A further tool for the assessment of the neonate’s circulation is blood pressure monitoring. Due to its feasibility, it is a common practice in the neonatal intensive care unit (NICU). Especially in compromised neonates during the first days after birth, blood pressure measurements are used to detect arterial hypotension, whereby thresholds are still controversially discussed [9,10,11,12,13,14,15]. During immediate transition, there are only a few observational studies describing blood pressure in preterm and term neonates [16].

Arterial blood pressure is dependent on systemic vascular resistance and cardiac output. Therefore, it determines tissue perfusion and oxygen delivery to the tissue. Tissue oxygenation can be monitored by near-infrared spectroscopy (NIRS) [17] The focus of tissue oxygenation measurements in neonates has been on cerebral tissue oxygenation measurement during the immediate transition [18,19] and at the NICU during the first week after birth [20]. Some studies in neonates have described that the combination of blood pressure measurements and cerebral NIRS monitoring might be a promising tool due to its potential to reveal information about cerebral autoregulation [21,22,23].

Therefore, the aim of the present qualitative systematic review was to identify publications on blood pressure monitoring in combination with cerebral tissue oxygenation monitoring with NIRS during the immediate transition and first week after birth in order to gain more information about cerebral autoregulation, resulting in improved treatment options and approaches in neonates.

## 2. Materials and Methods

### 2.1. The Search Strategy and Study Selection Criteria

Articles were identified using the stepwise approach specified in the Preferred Reporting Items for Systemic Reviews and Meta-Analyses Statement (PRISMA).

### 2.2. Search Strategy

A systematic search was performed on PubMed. In order to identify studies addressing blood pressure measurements, non-invasive or invasive in combination with cerebral tissue oxygenation measurements with NIRS during the immediate transition and first week after birth, the following search terms were used: infants/newborn/neonates, blood pressure/systolic/diastolic/mean/MAP/SAP/DAP, near-infrared spectroscopy, oxygenation/saturation/oxygen, and brain/cerebral. Additional studies were identified by a manual search of references in retrieved studies and reviews. Only human studies with combined blood pressure and cerebral tissue oxygenation monitoring during the immediate transition period after birth and the first week were included.

### 2.3. Study Selection

Articles identified following the literature review were evaluated by two authors (D.P., G.P.) for inclusion using the title and abstract. The full text was reviewed, resulting from remaining uncertainty regarding eligibility for inclusion. All data were analyzed qualitatively. Data extraction included the characterization of study types, patient demographics, methods, and results.

## 3. Results

The initial search detected 2200 articles on PubMed. Due to the rejection of studies that did not meet the prior mentioned criteria, 36 studies were identified, analyzing blood pressure and cerebral regional tissue oxygenation monitoring during the immediate transition and during the first week after birth (Figure 1).

Thirty-one studies described monitoring results in preterm neonates, and five studies described monitoring results in preterm and term neonates. There were no studies that conducted measurements on term neonates only.

Blood pressure was measured invasively with an indwelling catheter in 27 [21,23,24,25,26,27,28,29,30,31,32,33,34,35,36,37,38,39,40,41,42,43,44,45,46,47,48] studies and non-invasively with oscillometric measurements in 3 studies [22,49,50]. In five studies [51,52,53,54,55], both methods were combined, and in one study [56], the method was unclear. Cerebral oxygen saturation was measured in 19 studies with the NIRO 200, 300, or 500 (Hamamatsu Photonics, Hamamatsu-city, Japan) [23,26,27,28,29,30,32,34,35,36,37,40,42,44,45,48,51,53,56], in 10 studies with the INVOS 4100 or 5100 (Covidien, Medtronic, Minneapolis, MN, USA) [21,22,31,38,39,41,46,50,52,55], in 3 studies with the FORE-SIGHT (Casmed, Irvine, CA, USA) [24,25,47], in 2 studies with the cerebral tissue oxygenation Monitor 205 (Critikon, Tampa, FL, USA) [33,54], in 1 study with the Oxiplex TS.3.1 (ISS, Inc., Champaign, IL, USA) [49], and in 1 study with the NIRO 300 in combination with an INVOS 4100 [43] (Table 1, Table 2 and Table 3).

Two studies described the association of blood pressure values and cerebral NIRS during immediate transition in preterm and in term neonates [22,50] (Table 1). These studies showed intact cerebral autoregulation in term neonates and impaired cerebral autoregulation in moderate and late preterm neonates receiving respiratory support with significant associations between crSO2/cFTOE and MABP.

Twenty-six studies were conducted during the first 24 h [21,24,25,26,27,28,29,30,31,32,33,34,35,36,37,38,39,40,41,49,51,52,53,54,55,56] (Table 2). Eleven studies investigated physiological changes of blood pressure and cerebral tissue oxygenation [24,25,26,27,28,29,30,49,51,52,56]. A further six studies combined cerebral NIRS monitoring with blood pressure measurement to investigate cerebral autoregulation in stable and sick neonates [21,32,34,37,53,55]. Stable preterm neonates experienced reduced cerebral tissue oxygenation, perfusion, and cardiac output after birth, followed by an increase of all three parameters; however, cerebral autoregulation remained intact [28,29,49,51,52,56]. Besides, sick preterm neonates suffering from respiratory distress syndrome (RDS), hypotension, or sepsis had a higher prevalence of impaired cerebral autoregulation [21,25,32,34,37,53,55]. Another four studies focused on the treatment of hypotension and cerebral autoregulation [31,39,40,54]. They showed that cardio-circulatory treatment had a limited effect on cerebral autoregulation. The influence of impaired autoregulation on intraventricular hemorrhage (IVH), death, or abnormal neurodevelopmental outcome was demonstrated by five studies [33,35,36,38,41]

Eight studies were conducted 24 h after birth [23,42,43,44,45,46,47,48] (Table 3). Three studies examined the physiological changes in blood pressure, cerebral tissue oxygenation, and cerebral autoregulation [46,47,48], and a further two studies [43,45] investigated cerebral autoregulation in stable and sick neonates by combining cerebral tissue oxygenation with blood pressure measurement. Maintaining mean arterial blood pressure (MABP) within normal ranges reduces the duration of impaired cerebral autoregulation [46]. However, even clinically unremarkable preterm neonates below 32 weeks of gestational age still experience episodes of impaired cerebral autoregulation [47]. Risk factors for impaired cerebral autoregulation include a higher CRIB II Score [43,45]. The remaining three studies showed that impaired cerebral autoregulation increased the risk of IVH and abnormal neurodevelopmental outcomes [23,42,44].

## 4. Discussion

### 4.1. Immediate Transition

Studies within the immediate transition period analyzing blood pressure measurements in combination with cerebral tissue oxygenation measurements were scarce. There were only two observational studies available [22,50]. Baik et al. described that there is no correlation of cerebral oxygen saturation (crSO2) and cerebral fractional tissue oxygen extraction (cFTOE) with MABP in term neonates, suggesting intact cerebral autoregulation. These findings were in line with findings in animals by Helou et al. [48]. However, in moderate and late preterm neonates, Baik et al. [50] showed a significant correlation between cFTOE and MABP, whereas crSO2 and MABP did not correlate. Pfurtscheller et al. [22] showed in more detail that, only in moderate and late preterm neonates receiving respiratory support, both crSO2 and cFTOE were associated with MABP, indicating an impaired cerebral autoregulation in those compromised neonates.

### 4.2. First Day after Birth

Twenty-six studies were conducted during the first day after birth [21,24,25,26,27,28,29,30,31,32,33,34,35,36,37,38,39,40,41,49,51,52,53,54,55,56]. Of these, three studies described different mathematical and technical approaches to assess cerebral autoregulation [26,27,30]. Eight studies analyzed physiological changes of blood pressure and cerebral tissue oxygenation [24,25,28,29,30,49,51,56]. Naulaers et al. [56] demonstrated that, in preterm neonates, blood pressure, as well as the crSO2 values increased over the first three days. These findings in blood pressure were in line with normative blood pressure studies, which demonstrated an increase in MABP with an increase of postnatal age in days and with an increase in gestational age [16,57]. Concerning cerebral tissue oxygenation, Takami et al. [51] described a reduction after birth in stable preterm neonates in addition to a reduction in perfusion and cardiac output. These results seem to contradict their findings for blood pressure, since blood pressure did not correlate with cFTOE and crSO2, which suggests intact cerebral autoregulation. A further two studies were in line with the latter findings and showed that preterm neonates presenting a combination of low cardiac output and a normal systemic blood pressure were able to maintain cerebral and peripheral perfusion within normal ranges [28,29]. In contrast to those mentioned studies demonstrating intact cerebral autoregulation in preterm neonates [28,29,49,52], Gilmour et al. [24] demonstrated that preterm neonates could have episodes of impaired cerebral autoregulation in association with low arterial blood pressure. Their findings may be explained with the heterogeneity of their cohort, including stable and sick neonates. Bearing this in mind, Vesloulis et al. [25] showed that sick extremely low gestational age preterm neonates with a mean gestational age of 24 weeks had an autoregulatory immaturity, which led to a decrease in oxygen extraction with low blood pressure values.

Taking the above-mentioned studies into consideration, it seems that preterm neonates are able to have an intact cerebral autoregulation, but may lose this ability due to different risk factors. Six studies identified such risk factors for impaired cerebral autoregulation with blood pressure and cerebral NIRS measurements [21,32,34,37,53,55]. Hahn et al. [32] showed that inflammation in preterm neonates moderately influenced cerebral autoregulation in the first day after birth. However, it was unclear whether the impaired cerebral autoregulation was due to inflammation itself or due to arterial hypotension that exceeded cerebral autoregulatory capacity, caused by inflammation. These findings were in line with data coming from animal studies [58,59]. Furthermore, Lemmers et al. [21] demonstrated that neonates with RDS more frequently suffered from impaired cerebral autoregulation compared to neonates without RDS. These findings are supported by a non-NIRS study [60] demonstrating that CBF, measured by using the 133Xe clearance technique, varied with blood pressure, also suggesting an impaired cerebral autoregulation in preterm neonates with RDS. In addition to the previously mentioned risk factors for impaired cerebral autoregulation, birth weight and Clinical Risk Index for Babies (CRIB) Score were demonstrated to be risk factors as well, due to the influence on blood pressure variability, which exceeded cerebral autoregulatory capacity and led to fluctuations in cerebral tissue oxygenation [37]. Similar results concerning birth weight were shown by Baik et al. [61] by comparing cerebral NIRS data of intrauterine-growth-restricted (IUGR) neonates with appropriate for gestational-age neonates showing significantly higher crSO2 values and significantly lower cFTOE values in IUGR neonates during immediate transition. A further cause for impaired cerebral autoregulation is hypotension below the autoregulatory capacity [14]. This hazardous hypotension is commonly defined by MABP being below gestational age in NICU [62]. However, Binder et al. [55] demonstrated that, during borderline hypotension, cerebral autoregulation in preterm neonates was maintained within the first 24 h. These findings were in line with Dempsy’s multicenter HIP trial [13], where hypotensive preterm neonates with clinical evidence of good perfusion had equal cranial ultrasound outcomes as normotensive neonates, whereas neonates treated for low blood pressure were associated with adverse outcomes.

The treatment of arterial hypotension and its influence on cerebral autoregulation was the focus of four studies [31,39,40,54]. Bonestroo et al. [31] demonstrated in his study that any kind of hypotensive treatment did not cause a significant change in crSO2 and cFTOE. This is in line with Kooi et al. [39], who showed that cFTOE did not improve with volume expansion in hypotensive preterm neonates. On top of that, dopamine therapy was even associated with a decreased cerebral autoregulation in preterm neonates [40,54], and epinephrin or dopamine increased cerebral blood flow in sick preterm neonates.

Cerebral autoregulation assessment by blood pressure and cerebral tissue oxygenation measurements and the impact on outcome (intraventricular hemorrhage (IVH), death, or abnormal neurodevelopmental outcome) were addressed in five observational studies [33,35,36,38,41]. These studies demonstrated that impaired cerebral autoregulation within the first 24 h increased the risk for IVH [36,41] and abnormal neurodevelopmental outcome at 16 months [33]. This was in line with Chock et al. [38], who demonstrated that impaired cerebral autoregulation was associated with an increase of cerebral hemodynamic fluctuations, which increased the risk of death and IVH. Equal findings were reported by Da Costa et al. [34,35], who described impaired cerebral autoregulation in preterm neonates with IVH prior to and during the event.

### 4.3. Beyond the First Day after Birth

Eight studies combining blood pressure and cerebral tissue oxygenation measurements were conducted after 24 h up to 1 week after birth. [23,42,43,44,45,46,47,48]. One study showed different approaches to calculate cerebral autoregulation [48]. Two studies showed that, on the one side, with MABP maintained within normal ranges, the time with impaired cerebral autoregulation was reduced. On the other side, 40% of clinically unremarkable preterm neonates with a gestational age below 32 weeks showed episodes of impaired cerebral autoregulation within the first 72 h of life [46,47]. These findings are in accordance with observations during the first day after birth [24].

Two studies [43,45] assessed risk factors for impaired cerebral autoregulation, whereby the first showed that an increase in the CRIB II Score was associated with an impaired cerebral autoregulation [45]. This finding is again consistent with observations during the first day after birth [37]. The second study [43] demonstrated that investigating cerebral vascular reactivity with cerebral tissue oxygenation and heart rate measurements helped to identify neonates at risk. In this study, blood pressure and cerebral tissue oxygenation measurements showed no correlation, due to the described technical limitations.

Three studies investigated the influence of impaired cerebral autoregulation on IVH, death, or abnormal neurodevelopmental outcome with cerebral tissue oxygenation and blood pressure measurements [23,42,44]. The three studies [23,42,44] demonstrated similar results compared to studies that were performed during the first day after birth [33,35,36,38,41]. They concluded that the time with impaired cerebral autoregulation is associated with IVH, death, or abnormal neurodevelopmental outcome.

### 4.4. Limitations

First, one of the main limitations is the small cohorts in most of the included studies. The study populations were heterogeneous, in particular regarding the presence of risk factors. Thereby, most of the studies demonstrated that blood pressure measurement in combination with cerebral tissue oxygenation monitoring is a feasible method for bedside monitoring of cerebral autoregulation.

Second, studies were performed using different blood pressure measurement methods (invasive and non-invasive) and different NIRS devices.

Third, different methods were used to measure cerebral autoregulation, and different thresholds were set for defining impaired autoregulation.

## 5. Conclusions

Interpreting arterial blood pressure measurements and making therapeutic decisions can be challenging in clinical practice. The use of cerebral tissue oxygenation provides a promising approach for establishing blood pressure targets that preserve cerebral autoregulation and prevent pressure-passive cerebral perfusion. Integrating blood pressure monitoring with cerebral tissue oxygenation measurements provides the potential to identify more effectively interventions for improving neurodevelopmental outcomes in high-risk patients. This approach has significant implications for enhancing clinical practice and ultimately improving patient outcomes.

## Figures and Tables

**Figure 1 children-10-01304-f001:**
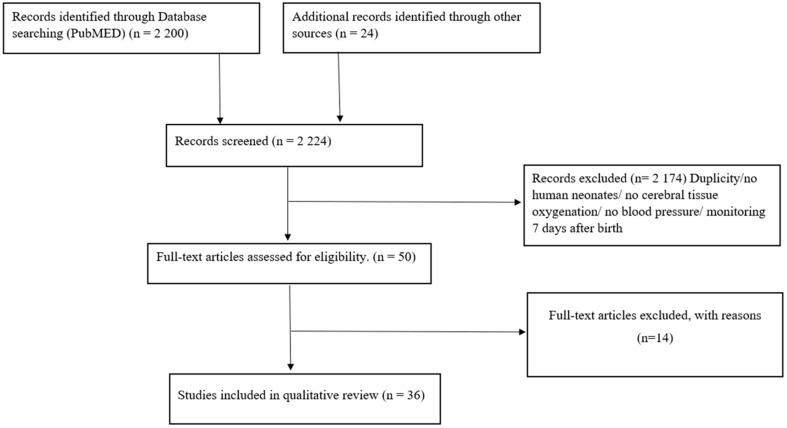
Selection of papers.

**Table 1 children-10-01304-t001:** Cerebral tissue oxygenation measurement in combination with arterial blood pressure measurement during immediate transition.

Author (Reference) (Year) of Publication	Neonates, *n*	Gestational Age, Weeks	NIRS Device	Arterial Blood Pressure	Duration	Initiation	Study Aim	Main Results Concerning BP and NIRS
Baik [50] (2017)	Term *n* = 292, preterm *n* = 186	Term 38.9 ± 0.8, preterm 31.0 ± 3.5	INVOS 5100c	MABP (oscillometric)	15 min	0–1 min	Impact of MABP on the cerebral regional oxygen saturation	In preterm neonates, MABP correlated negatively with cFTOE
Pfurtscheller [22] (2022)	Preterm *n* = 47	34.4 ± 1.6 (resp. support, *n* = 25) 34.5 ± 1.5 (stable, *n* = 22)	INVOS 5100c	MABP (oscillometric)	15 min	0–1 min	Impact of MABP and NIRS parameters in compromised neonates	In compromised preterm neonates, MABP correlated negatively with cFTOE and positively with crSO2

BP, blood pressure; cFTOE, cerebral fractional tissue oxygen extraction; crSO2, cerebral oxygen saturation; MABP, mean arterial blood pressure; NIRS, near-infrared spectroscopy; resp., respiratory.

**Table 2 children-10-01304-t002:** Cerebral tissue oxygenation measurement in combination with blood pressure measurement after immediate transition within the first 24 h after birth.

Author (Reference) (Year) of Publication	Neonates, *n*	Gestational Age, Weeks	NIRS Device	Arterial Blood Pressure Evaluation	Duration	Initiation	Study Aim	Main Results Concerning BP and NIRS
Naulaers [56] (2002)	Preterm *n* = 15	28.0 (25.0–30.0)	NIRO 300	n.a.	48 h	<6.0 h	To describe normal values of cTOI in premature infants	cTOI, MABP, and CBF increased in the first 3 days in preterm neonates
Pellicer [54] (2005)	Preterm *n* = 59	28.3 ± 2.3	Critikon	MABP (oscillometric and invasive)	80 min	5.3 ± 3.7 h	Effect of two catecholamines on brain hemodynamics in LBW neonates	Epinephrin and dopamine increased BP, CBF, and HbD, whereas cerebral circulation is still pressure passive
Lemmers [21] (2006)	Preterm *n* = 83	26.6 ± 1.32 (with RDS, *n* = 18) 29.3 ± 1.74 (without RDS, *n* = 20)	INVOS 4100	MABP (invasive)	72 h	1.0–2.0 h	The influence of RDS on arterial blood pressure in preterm neonates with and without RDS	RDS neonates showed impaired CA with positive MABP-crSO2 and negative MABP–cFTOE correlations
Victor [29] (2006)	Preterm *n* = 40	27.0 (23.0–30.0)	NIRO 500	MABP (invasive)	96 h	<24.0 h	Association between cardiocirculatory values and cerebral oxygenation	Stable very premature neonates showed intact CA without correlation of MABP-cFTOE and CO-cFTOE
Victor [28] (2006)	Preterm *n* = 35	27.0 (24.0–34.0)	NIRO 500	MABP (invasive) and echo	96 h	<24.0 h	Association between cardiocirculatory values and cerebral monitoring	aEEG and cFTOE maintained normal above MABP of 23 mmHg
O’Leary [36] (2009)	Preterm *n* = 88	26.0 (23.0–30.0)	NIRO 500	MABP (invasive)	96 h	11.0 h	Association between CA and outcome	MAP-HbD gain reflecting cerebral pressure passivity was associated with IVH or PVL
Hahn [30] (2010)	Preterm *n* = 22	27.5 (24.0–29.0)	NIRO 300	MABP (invasive)	1.3–3.7 h	17.4 h	Increasing precision of coherence analysis by adding MABP	CA measurements took hours and can be improved by adding MABP
Takami [51] (2010)	Preterm *n* = 16	25.2 ± 1.6	NIRO 200 NIRO 300	MABP (oscillometric and invasive) and echo	72 h	3.0–6.0 h	Detailed analyses of cerebral oxygenation and cardiac function	cTOI decreased initially, then increased, while FTOE showed the opposite pattern; MABP increased gradually
Bonestroo [31] (2011)	Preterm *n* = 142	30.0 (26.0–31.6) (volume, *n* = 33) (control 1, *n* = 33) 29.4 (25.9–31.6) (dopamine, *n* = 38) (control 2, *n* = 38)	INVOS 4100–5100	MABP (invasive)	1 h	15 min before treatment	Effect of volume expansion and dopamine in hypotensive preterm neonates	No significant changed in rScO2 and cFTOE
Gilmore [24] (2011)	Preterm *n* = 23	26 ± 1	Foresight	MABP (invasive)	24–96 h	14.4 ± 14.4 h	Relationship between CA and blood pressure	Correlation between MABP and impaired CA
Hahn [32] (2012)	Preterm *n* = 60	27 ± 1	NIRO 300	MABP (invasive)	2.3 h	2.3 ± 0.5 h	Neonates with inflammation and CA	Impairment of CA measured with OI worsened with lower MABP
Wong [37] (2012)	Preterm *n* = 32	26.3 ± 1.5	NIRO 200	MABP (invasive)	57.0 ± 5.9 h	12 ± 5.8 h	Relationship between cerebral autoregulatory capacity and blood pressure	Sick infants exhibited blood pressure-dependent variations in crSO2
Alderliesten [41] (2013)	Preterm *n* = 90	27.9 (26.2–30.0) (with IVH, *n* = 30) 27.5 (25.4–31.0) (without IVH, *n* = 60)	INVOS 4100–5100	MABP (invasive)	24 h after IVH	21.0 h	Association between CA and IVH	IVH infants exhibited increased crSO2, decreased cFTOE, and passive brain perfusion indicated by MABP–crSO2 correlation
Kooi [39] (2013)	Preterm *n* = 14	27.6 (25.0–28.7)	INVOS 5100C	MABP (invasive)	1 h after volume therapy	16.8 h	Effect of volume therapy in hypotensive neonates	Volume did not improve cFTOE in preterm neonates
Eriksen [40] (2014)	Preterm *n* = 60	26.2 ± 1.5 (dopamine, *n* = 13) 26.7 ± 1.2 (no dopamine, *n* = 47)	NIRO 300	MABP (invasive)	2.3 ± 0.5 h	18 ± 9.4 h	Effect of dopamine therapy in terms of CA	Dopamine therapy was associated with decreased CA
Riera [27] (2014)	Preterm *n* = 54	27 ± 2	NIRO 200NX	MABP (invasive)	9.5 h	<24.0 h	To identify impaired hemodynamics	BIAR COH (a specific time–frequency analysis consisting of MABP and TOI) identified cerebral hypoperfusion
Binder-Heschl [55] (2015)	Preterm *n* = 46	33.4 ± 1.9 (hypotensive, *n* = 17) 33.3 ± 1.3 (normotensive, *n* = 29)	INVOS 5100	MABP (oscillometric and invasive) and echo	24 h	<6.0 h	CA during hypotension	There were no significant differences in mean 24-h crSO2 and cFTOE between hypotensive and normotensive neonates
Demel [49] (2015)	Term *n* = 7, Preterm *n* = 16	39.9 (37.0–40.2) (term, *n* = 7) 34.0 (32.2–35.6) (preterm, *n* = 16)	Oxiplex TS 3.1	MABP (oscillometric)	72 h	7.0–11.0 h term 1.5–2.0 h preterm	Feasibility of NIRS and Doppler sonography	Measurements of crSO2 using frequency domain NIRS was feasible
Eriksen [26] (2015)	Preterm *n* = 60	27 ± 1	NIRO 300	MABP (invasive)	2.3 ± 0.5 h	18.0 ± 9.4 h	Comparison of two conventional methods used to describe CA	Time domain analysis using TOI and MABP appeared more robust in describing CA
Stammwitz [33] (2016)	Preterm *n* = 31	27.3 (26.0–32.0)	Critikon	MABP (invasive)	68–76 h	<6.0 h	Association between CA and outcome	Higher variability of TOI was associated with IVH and death
Vesoulis [25] (2017)	Preterm *n* = 68	25.5 ± 1.3	Foresight	MABP (invasive)	72 h	17.8 ± 9.7 h	Evaluation of the interaction between BP, changes in oxygen extraction, and maturity	In extreme preterm neonates, MABP and cFTOE showed a positive correlation, indicating immature autoregulation
Da Costa [34] (2018)	Preterm *n* = 44	25.0 (23.0–27.0)	NIRO 200NX	MABP (invasive)	24 h	3.1–12.6 h	To define optimal MABP using NIRS	Optimal MABP gained by TOI and HR identified risk patients
Pichler [53] (2018)	Preterm *n* = 98	33.1 (32.0–34.0) (with NIRS, *n* = 49) 33.4 (32.3–34.3) (without NIRS, *n* = 49)	NIRO 200NX	MABP (oscillometer and invasive)	48 h	2.0 (1.5–3.5) h (with NIRS) 2.5 (2.0–4.0) h (without NIRS)	Reduction of hypotensive episodes by using NIRS	cTOI measurements led to a non-significant reduction in arterial hypotension
Da Costa [35] (2019)	Preterm *n* = 43	25.7 (23.6–31.0)	NIRO 200NX	MABP (invasive) and echo	48 h	6.0 h	Association of MABP and IVH in preterm neonates	crSO2 was lower in neonates with IVH before and during the event
Bruckner [52] (2020)	Term *n* = 13, preterm *n* = 47	34.0 (33.0–35.0) (whole cohort)	INVOS 5100	MABP (oscillometric and invasive) and echo	24 h	4.0–6.0 h	Association between cardiac function and crSO2	In stable term and preterm neonates, crSO2 and cFTOE did not correlate with CO
Chock [38] (2020)	Preterm *n* = 103	26.2 ± 1.7	INVOS 5100C	MABP (invasive)	96 h	8.0–21.0 h	Association between CA and outcome	MABP and crSO2 correlated in neonates with adverse outcome

BP, blood pressure; CBF, cerebral blood flow; CA, cerebral autoregulation; cFTOE, cerebral fractional tissue oxygen extraction; CO, cardiac output; crSO2, cerebral oxygen saturation; cTOI, cerebral tissue oxygenation index; EEG, electroencephalogram; HbD, deoxygenated hemoglobin; HR, heart rate; IVH, intraventricular hemorrhage; LBW, low birth weight; MABP, mean arterial blood pressure; NIRS, near-infrared spectroscopy; OI, oxygenation index; PVL, periventricular leukomalacia.

**Table 3 children-10-01304-t003:** Cerebral tissue oxygenation measurement in combination with blood pressure measurement after 24 h up to 1 week after birth.

Author (Reference) (Year) of Publication	Neonates, *n*	Gestational Age, Weeks	NIRS Device	Arterial Blood Pressure Evaluation	Duration	Initiation	Study Aim	Main Results Concerning BP and NIRS
Tsuji [23] (2000)	Preterm *n* = 32	26 (23.0–31.0)	NIRO 500	MABP (invasive)	30 min	<72 h	Association between CA and outcome	Concordant changes in HbD and MABP suggest impaired cerebrovascular function
Wong [42] (2008)	Preterm *n* = 24	26 ± 2	NIRO 300	MABP (invasive)	3 h	28 h	Association between CA and outcome	High coherence between MABP and cTOI indicates impaired CA in sick preterm neonates
De Smet [48] (2009)	Term and preterm *n* = 20	28.7 (24.0–39.0)	NIRO 300	MABP (invasive)	1.5–23.5 h	<72 h	To assess whether cTOI may replace HbD for measuring CA	cTOI and HbD showed similar results; both may be used for calculating CA
Caicedo [43] (2011)	Preterm *n* = 53	29 ± 2	INVOS 4100 and NIRO 300	MABP (invasive)	6–70 h	24–72 h	To assess whether cTOI and crSO2 may replace HbD for measuring CA	cTOI, crSO2, and HbD showed similar results; all three may be used for calculating CA
Zhang [44] (2011)	Preterm *n* = 17	26.4 (24.0–29.0)	NIRO 300	MABP (invasive)	72 h	24–72 h	Association between CA and outcome	Neonates with IVH showed higher TOI, lower cFTOE, and reduced coherence between MABP and HbD
Mitra [45] (2014)	Preterm *n* = 31	26.1 (23.7–32.6)	NIRO 200NX	MABP (invasive)	2 h	48 h	Association between cardio-circulatory values and CBF in sick preterm neonates	cTOI and HR, reflecting cerebrovascular reactivity, showed a correlation with MABP
Verhagen [46] (2014)	Preterm *n* = 25	29.1 (25.4–31.7)	INVOS 4100–5100	MABP (invasive)	24 h	<72 h	Association between clinical variables and CA	Negative correlation between MABP and cFTOE suggests the absence of CA
Traub [47] (2021)	Preterm *n* = 17	26.5 (23.0–33.2)	Foresight	MABP (invasive)	24 h	88.8 h	To determine whether NIRS helps to identify neonates at risk	Neonates maintain intact CA within normal MABP ranges

CA, cerebral autoregulation; CBF, cerebral blood flow; cFTOE, cerebral fractional tissue oxygen extraction; crSO2, cerebral oxygen saturation; cTOI, cerebral tissue oxygenation index; HbD, deoxygenated hemoglobin; HR, heart rate; IVH, intraventricular hemorrhage; MABP, mean arterial blood pressure; NIRS, near-infrared spectroscopy.

## Data Availability

No new data were created nor analyzed in this study. Data sharing is not applicable to this article.

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
