# Peer review of "Insights into Neonatal Cerebral Autoregulation by Blood Pressure Monitoring and Cerebral Tissue Oxygenation: A Qualitative Systematic Review"

_children, 2023, doi:10.3390/children10081304_

Round 1
Reviewer 1 Report
The manuscript tackles an important issue of neonatal care that can provide insightful evidence of hemodynamic impairment in sick neonates. However, the writing manner should be improved, as follows:
Both for the Abstract and the main body of the manuscript - The Results section is insufficient. It should include the main findings of the discussed articles. This means information from rows 176-180, 182-187, 190-194, 198-207, 209-212, 215-220, 222-223, 243-260, 263-283, pertains in the Results section.
Abstract – please rephrase the first sentence on row 45 – it has a very German turn of phrase.
On row 97, please add “t” at the end of “Statemen” and also add PRISMA as the abbreviation.
The Figures and Tables the authors included are waaay too small, this meaning they are unreadable. At least for peer-review’s sake, all tables should be provided in the same manner as Table 3.
Figure 1 - apart from the size, the title is missing.
On rows 124-125, please provide references for the respective studies. On row 126, please correct „method”.
Table 1 – the title should start with a capital „C”.
Please rephrase the information on row 156, to better differentiate it from that on row 142.
On row 232, the abbreviation IUGR is unnecessary, as it is used only that one time.
The Limitations section need not be a distinct section, it can be a subtitle in the Discussions section. Also, the information on rows 294-295 should be deleted.
As mentioned above
Author Response
The manuscript tackles an important issue of neonatal care that can provide insightful evidence of hemodynamic impairment in sick neonates. However, the writing manner should be improved, as follows:
R1: “Both for the Abstract and the main body of the manuscript - The Results section is insufficient. It should include the main findings of the discussed articles. This means information from rows 176-180, 182-187, 190-194, 198-207, 209-212, 215-220, 222-223, 243-260, 263-283, pertains in the Results section. “
Answer: Changes were done accordingly in the Abstract
Page 1 Line 37 “Thirty-one studies focused on preterm neonates, while five included preterm and term neonates. In stable term neonates intact cerebral autoregulation was shown by combining cerebral tissue oxygenation and blood pressure during immediate transition, while impaired autoregulation was observed in preterm neonates with respiratory support. Within the first 24 hours, stable preterm neonates had reduced cerebral tissue oxygenation with intact cerebral autoregulation, while sick neonates showed a higher prevalence of impaired autoregulation. Further cardio-circulatory treatment had limited effect on cerebral autoregulation. Impaired autoregulation, with dependency on blood pressure and cerebral tissue oxygenation, increased the risk of intraventricular hemorrhage and abnormal neurodevelopmental outcomes..”
Changes were done accordingly in the results section:
- Page 4, Line 135-138 “These studies showed intact cerebral autoregulation in term neonates and impaired cerebral autoregulation in moderate and late preterm neonates receiving respiratory support with significant associations between crSO2/cFTOE, and MABP.”
- Page 6, Line 152-161 “Stable preterm neonates experienced reduced cerebral tissue oxygenation, perfusion, and cardiac output after birth, followed by an increase of all three parameters, however cerebral autoregulation remained intact (24,25, 30–32, 34). Besides sick preterm neonates suffering from respiratory distress syndrome (RDS), hypotension or sepsis had a higher prevalence of impaired cerebral autoregulation (21, 27, 37, 38, 40, 42, 45). Another four studies focused on the treatment of hypotension and cerebral autoregulation (36, 39, 47, 48). They showed that cardio-circulatory treatment had limited effect on cerebral autoregulation. Influence of impaired autoregulation on intraventricular hemorrhage (IVH), death, or abnormal neurodevelopmental outcome has been demonstrated by five studies (41, 43, 44, 46, 49).”
- Page 11, Line175-192 “Three studies examined the physiological changes in blood pressure, cerebral tissue oxygenation, and cerebral autoregulation (54-56) and further two studies (51, 53) investigated cerebral autoregulation in stable and sick neonates by combining cerebral tissue oxygenation with blood pressure measurement. Maintaining mean arterial blood pressure (MABP) within normal ranges reduces the duration of impaired cerebral autoregulation (54). However, even clinically unremarkable preterm neonates below 32 weeks of gestational age still experience episodes of impaired cerebral autoregulation (55). Risk factors for impaired cerebral autoregulation include a higher CRIB II Score (51, 53). The remaining three studies explored the impact of impaired cerebral autoregulation on intraventricular hemorrhage (IVH), death, or abnormal neurodevelopmental outcome (23, 50, 52).”
R1 Abstract – please rephrase the first sentence on row 45 – it has a very German turn of phrase.
Answer: The Abstract has been rewritten
R1 On row 97, please add “t” at the end of “Statemen” and also add PRISMA as the abbreviation.
Answer: Page 3, Line 95 the typo was corrected as well as the abbreviation added according to the Reviewer
R1 The Figures and Tables the authors included are waaay too small, this meaning they are unreadable. At least for peer-review’s sake, all tables should be provided in the same manner as Table 3.
Answer: The reviewer is right, and this issue is solved accordingly
R1 Figure 1 - apart from the size, the title is missing.
Answer: The title was added Page 3 Line 117 “Figure 1. Selection of papers”
R1 On rows 124-125, please provide references for the respective studies.
Answer the following references were added as recommended.
Page 4 Line 123 “Blood pressure was measured invasively with an indwelling catheter in 27 (21, 23 26–31, 33, 36, 37, 41–56) studies and non-invasively with oscillometric measurements in three studies (22, 32, 35). In five studies (24, 34, 38, 39, 40), both methods were combined, and in one study (25) the …”
R1 On Rows 126, please correct „method”.
Answer: Page 4 Line 126 method was corrected as recommended
Table 1 – the title should start with a capital „C”.
Answer: Page 4 Line 148 The title was corrected accordingly
R1: Please rephrase the information on row 156, to better differentiate it from that on row 142.
Answer: Page 11 Line 184 “The remaining three studies showed that impaired cerebral autoregulation increased the risk of IVH and abnormal neurodevelopmental outcomes (23, 50, 52).”
R1 On row 232, the abbreviation IUGR is unnecessary, as it is used only that one time.
Answer: The abbreviation is used again in Row 260 therefore it is maintained.
R1: The Limitations section need not be a distinct section; it can be a subtitle in the Discussions section.
Answer: Page Changes were done accordingly.
R1 Also, the information on rows 294-295 should be deleted.
Answer: the information on rows 294-295 were deleted accordingly

Reviewer 2 Report
Dear authors,
Neonatal cerebral autoregulation is a fascinating subject and worth studying, that's why I congratulate you for the initiative to systematize the studies carried out in this direction. Regarding your manuscript, I have some comments and recommendations.
Abstract: I think that the Abstract is the essence of a manuscript and it should highlight the importance of the chosen subject and the results that emerge after a rigorous research of the literature. Your abstract lists the articles found on the topic studied and loses sight of the results of the respective studies. I would recommend redoing the Abstract in a more descriptive manner.
Materials and Methods - A systematic search should include more medical databases, not only PubMed
Results: -Figure 1 is very difficult to read, it must be enlarged. Same for tables.
Line 294-295: - please revise the phrase
Conclusion: Please be more specific: what is the main conclusion of the study?
Author Response
Reviewer 2
Dear authors,
Neonatal cerebral autoregulation is a fascinating subject and worth studying, that's why I congratulate you for the initiative to systematize the studies carried out in this direction. Regarding your manuscript, I have some comments and recommendations.
R2: Abstract: I think that the Abstract is the essence of a manuscript and it should highlight the importance of the chosen subject and the results that emerge after a rigorous research of the literature. Your abstract lists the articles found on the topic studied and loses sight of the results of the respective studies. I would recommend redoing the Abstract in a more descriptive manner.
Answer: The Abstract was redon according to the Reviewer.
Page 1 Line 25 “Abstract:
Objective:
Aim of this qualitative systematic review was to identify publications on blood pressure moni-toring in combination with cerebral tissue oxygenation monitoring during the first week after birth focusing on cerebral autoregulation.
Methods:
A systematic search was performed on PubMed. The following search terms were used: in-fants/newborn/neonates, blood pressure/systolic/diastolic/mean/MAP/SAP/DAP, near-infrared-spectroscopy, oxygenation/saturation/oxygen and brain/cerebral. Additional stud-ies were identified by a manual search of references in retrieved studies and reviews. Only human studies were included.
Results:
Thirty-one studies focused on preterm neonates, while five included preterm and term neonates. In stable term neonates intact cerebral autoregulation was shown by combining cerebral tissue oxygenation and blood pressure during immediate transition, while impaired autoregulation was observed in preterm neonates with respiratory support. Within the first 24 hours, stable preterm neonates had reduced cerebral tissue oxygenation with intact cerebral autoregulation, while sick neonates showed a higher prevalence of impaired autoregulation. Further cardio-circulatory treatment had limited effect on cerebral autoregulation. Impaired autoregulation, with depend-ency on blood pressure and cerebral tissue oxygenation, increased the risk of intraventricular hemorrhage and abnormal neurodevelopmental outcomes.
Conclusion:
Integrating blood pressure monitoring with cerebral tissue oxygenation measurements has the potential to improve treatment decisions and optimizes neurodevelopmental outcomes in high-risk neonates.”
R2: Materials and Methods - A systematic search should include more medical databases, not only PubMed
Answer: The Reviewer might be right. However, PubMed is one of the largest and most widely used medical databases, covering most biomedical literature. Furthermore, additional studies were identified by a manual search of references in retrieved studies and reviews. Therefore, we are convinced that including further databases would not helped to identify further relevant studies.
Results: -Figure 1 is very difficult to read, it must be enlarged. Same for tables.
Answer: Tables and Figures were adjusted accordingly
Line 294-295: - please revise the Phrase.
Answer: the phrase on rows 294-295 were deleted accordingly
Conclusion: Please be more specific: what is the main conclusion of the study?
Answer: The Conclusion was rewritten to be more specific
Page 16 Line 321 “Interpreting arterial blood pressure measurements and making therapeutic decisions can be challenging in clinical practice. The use of cerebral tissue oxygenation provides a promising approach for establishing blood pressure targets that preserve cerebral autoregulation and prevent pressure-passive cerebral perfusion. Integrating blood pressure monitoring with cerebral tissue oxygenation measurements provides the potential to identify more effectively interventions for improving neurodevelopmental outcomes in high-risk patients. This approach has significant implications for enhancing clinical practice and ultimately improving patient outcomes.”

Round 2
Reviewer 1 Report
I appreciate the authors’ willingness to make the necessary changes to the manuscript and I feel that this has much improved the quality in which the research is presented.
However, I wonder which is the final version of the manuscript, as there are three versions listed in the Author Center: a PDF version under manuscript, and two MS Word versions, one titled “supplementary” and the other “coverletter”. They seem fairly similar, but I haven’t compared them word for word.
There is one small comment I have to make, for Table 3 (page 15 of the PDF version) – the choice of the word “calculation” for cerebral autoregulation seems inappropriate – please replace with “assessment” or “evaluation”.
Reviewer 2 Report
Dear authors,
I found your manuscript improved and suitable for publication.
Kind regards,